# Aβ-Induced Alterations in Membrane Lipids Occur before Synaptic Loss Appears

**DOI:** 10.3390/ijms23042300

**Published:** 2022-02-19

**Authors:** Michiel Van Bulck, Nicola Brandt, Ralf A. Claus, Markus Gräler, Anja U. Bräuer

**Affiliations:** 1Research Group Anatomy, School for Medicine and Health Science, Carl von Ossietzky University Oldenburg, 26129 Oldenburg, Germany; nicola.brandt@uni-oldenburg.de; 2Department of Experimental Models of Human Disease, Networked Center of Biomedical Research on Neurodegenerative Diseases (CIBERNED), Institute for Biomedical Research A. Sols (CSIC-UAM), 28029 Madrid, Spain; 3Department of Anaesthesiology and Intensive Care Medicine Center for Molecular Biomedicine(CMB), Jena University Hospital, 07745 Jena, Germany; ralf.claus@med.uni-jena.de (R.A.C.); markus.graeler@med.uni-jena.de (M.G.); 4Centre for Sepsis Control and Care (CSCC), Jena University Hospital, 07745 Jena, Germany; 5Research Centre for Neurosensory Science, Carl von Ossietzky University Oldenburg, 26129 Oldenburg, Germany

**Keywords:** neurones, glia cells, synapses, Aβ species, membrane lipids, Alzheimers disease, cell-communication

## Abstract

Loss of active synapses and alterations in membrane lipids are crucial events in physiological aging as well as in neurodegenerative disorders. Both are related to the abnormal aggregation of amyloid-beta (Aβ) species, generally known as amyloidosis. There are two major known human Aβ species: Aβ_(1–40)_ and Aβ_(1–42)_. However, which of these species have more influence on active synapses and membrane lipids is still poorly understood. Additionally, the time-dependent effect of Aβ species on alterations in membrane lipids of hippocampal neurones and glial cells remains unknown. Therefore, our study contributes to a better understanding of the role of Aβ species in the loss of active synapses and the dysregulation of membrane lipids in vitro. We showed that Aβ_(1–40)_ or Aβ_(1–42)_ treatment influences membrane lipids before synaptic loss appears and that the loss of active synapses is not dependent on the Aβ species. Our lipidomic data analysis showed early changes in specific lipid classes such as sphingolipid and glycerophospholipid neurones. Our results underscore the potential role of lipids as a possible early diagnostic biomarker in amyloidosis-related disorders.

## 1. Introduction

Amyloidosis is used as an umbrella term for rare, serious diseases caused by the deposit of misfolded proteins. In brain tissue, it is characterised by the accumulation of amyloid-beta (Aβ), such as occurs in Alzheimer’s disease (AD) [1]. Aβ species are products of a proteolytic cleavage, generated from amyloid precursor protein (APP) by α- or β-secretase and γ-secretase activity, and their characteristics have been extensively reviewed [1,2,3]. However, the presumed neurotoxic effects of the major Aβ species, Aβ_(1–40)_ and Aβ_(1–42)_, under pathological and physiological conditions remain unclear. So far, the molecular and cellular mechanism of Aβ species and their impact on the loss of synaptic sites, as well as on the changes of membrane lipids of brain cells, is poorly understood. 

APP has been shown to play a pivotal role in synaptic and neural plasticity [4]. In vitro and in vivo studies have demonstrated that soluble Aβ species accumulate at the synaptic sites, resulting in disrupted synaptic plasticity and long-term potential [5,6,7,8]. However, Aβ_(1–42)_ is thought to be more aggregation-prone compared to Aβ_(1–40)_. The aggregation status of Aβ species in senile plaques in AD is strongly regulated by time and their aggregation affinity [9,10]. Therefore, as has been demonstrated in vitro and in vivo, Aβ species have different influences on the pre- and postsynaptic densities dependent on the Aβ concentrations, the chemical structure of Aβ (α-helices or β-sheets), the time of Aβ treatment, and the Aβ aggregation status [2,11,12,13,14,15,16,17,18,19,20,21]. Previously, it was also shown that glial cells are involved in Aβ-induced inflammatory responses and play an important role in Aβ clearance and degradation [22]. In addition, glial activation itself could play a protective role against Aβ-induced toxicity on neurones [22].

APP and the cleavage products Aβ_(1–40)_ and Aβ_(1–42)_ have different influences on lipid homeostasis [23]. Major molecular targets for Aβ in the cholesterol and sphingolipid metabolic pathways are 3-*Hydroxy-3-methylglutaryl**-**coenzyme A* (HMG-CoA) reductase and sphingomyelinases (nSMase) [23]. Aβ_(1–42)_ activates nSMase, whereas Aβ_(1–40)_ suppresses the activity of HMG-CoA reductase, resulting in decreased cholesterol levels. Sphingomyelins (SM) reduce γ-secretase activity, thus reducing Aβ levels; cholesterol, on the contrary, induces γ-secretase activity and is responsible for elevated Aβ levels [23,24]. Inhibition of HMG-CoA reductase is responsible for a reduction of intracellular as well as extracellular Aβ_(1–40)_ and Aβ_(1–42)_ peptides, resulting in elevated levels of cholesterol [25]. Sphingosine-1-phosphate (So1P) has been shown to be protective for neuronal cells, whereas ceramide promotes Aβ biogenesis by influencing the β-secretase of APP [26]. Ceramides also interfere in the control of many cellular processes, influencing, e.g., Aβ aggregation in physiological and pathological aging processes [27]. Phosphatidylcholines can alter the Aβ_(1–40)_ mediated aggregation, depending on the thickness of the lipid membrane [28]. Charged phospholipid bilayers consisting of phosphatidylcholines and phosphoglycerol showed an increase in Aβ_(1–40)_ fibril formation [29]. 

In this study, we investigated the direct effect and time-dependent influence of two major human Aβ species on primary hippocampal neurones and on glial cells. Hence, we examined whether the loss of active synapses in primary hippocampal neurones is Aβ-species- and time-dependent in vitro. 

Our results show for the first time a detailed lipid profiling in two different types of brain cells, primary hippocampal neurones and glial cells, after 3 h (3 h) and 12 h (12 h) of Aβ_(1–40)_ and Aβ_(1–42)_ treatment. We established that both Aβ species cause changes in the membrane lipids of primary hippocampal neurones and glial cells after 3 h and 12 h of treatment. Interestingly, Aβ-induced alterations in membrane lipids occurred prior to the loss of active synapses.

## 2. Material and Methods

### 2.1. Animals

For all experiments, timed-pregnant and postnatal mice were obtained from the central animal facility of the Carl von Ossietzky Universität Oldenburg. The primary hippocampal neurones were derived from C57 BL/6 mouse embryos at embryonic stage 18 (E18). These experiments were carried out in accordance with the institutional guidelines for animal welfare and approved by the “Niedersächsisches Landesamt für Verbraucherschutz und Lebensmittelsicherheit” (33.19-45502-04-18/2766). The glial cells were obtained from postnatal day 2 (P2) mouse pups in accordance with the institutional guidelines of German animal welfare (§4) for the use of laboratory animals at the Carl von Ossietzky Universität Oldenburg. 

### 2.2. Primary Hippocampal Neurone Cultures

Primary hippocampal neurones were prepared from E18 mouse embryos. The hippocampi were dissected and pooled in a 15 mL falcon tube with HBSS (1X, Phenol red, Thermo Fisher Scientific, Waltham, MA, USA) and stored on ice. Afterward, they were washed twice in HBSS and incubated in 0.25% trypsin (Gibco, Thermo Fisher Scientific) for 15 min at 37 °C. The supernatant was then carefully removed and replaced by plating media, containing MEM (Gibco, Thermo Fisher Scientific), supplemented with 0.6% D-(+)-glucose (≥99.5, Sigma-Aldrich, Taufkirchen, Germany), 10% horse serum (Gibco, Thermo Fisher Scientific) and 100 U/mL penicillin with 100 µg/mL streptomycin (Pan-Biotech, Aidenbach, Germany). They were subsequently triturated with a fire-polished glass pipette until neurones were dissociated. Before plating the cells, culture plates and glass coverslips (Marienfeld 12 mm, VWR, Darmstadt, Germany) were coated with 0.2 mg/mL poly-L-lysine (PLL, P2636-100MG, Sigma-Aldrich) in *ortho*-boric acid (VWR chemicals, Darmstadt, Germany) buffer (pH 8.5) overnight, followed by rinses with cell culture water (USP WFI, Lonza, Biozym, Germany). The dissociated neurones were filtered through a 40 µm cell strainer (Greiner Bio-one, Frickenhausen, Germany), and for immunofluorescence, ~3 × 10^4^ neurones were plated into 24-well plates (TPP^®^ Faust lab Science, Klettgau, Germany). For high-performance thin-layer chromatography (HPTLC) and tandem mass spectrometry (MS/MS) analysis, ~1 × 10^6^ neurones were seeded into 60 mm-diameter petri dishes (Nunclon^TM^ Delta surface, Thermo Fisher Scientific) containing plating media. Three to four hours after plating, cells were washed twice with 1 x PBS (phosphate-buffered Saline, Thermo Fisher Scientific), incubated, and maintained in Neurobasal A media (Gibco, Thermo Fisher Scientific) supplemented with 0.25% L-glutamine (200 mM, Sigma Aldrich, Omnilab, Bremen, Germany), 2% B27-supplement (50X, Thermo Fisher Scientific) and 100 U/mL penicillin with 100 µg/mL streptomycin (Pan-Biotech). Hippocampal neurones used for synaptic study and lipid analysis were maintained for 12 days in vitro (DIV) at 37 °C, 5% CO_2_.

### 2.3. Primary Glial Cell Cultures

Glial cell cultures were prepared from cortical tissue of P2 mouse pups. The brains were dissected and washed twice in HBSS (1X, Phenol red, Thermo Fisher Scientific). After removing the meninges from the brains, cortical tissue was isolated, washed twice in HBSS and incubated with 0.05% trypsin-EDTA (Gibco, Thermo Fisher Scientific) for 12 min at 37 °C. The supernatant was then carefully removed and replaced by plating media, consisting of DMEM (high glucose with L-glutamine, 4500 mg/L D-glucose, 110 mg/L sodium pyruvate, Gibco, Thermo Fisher Scientific) supplemented with 10% FBS (Pan-Biotech), 10% HS (Pan-Biotech) and 100 U/mL penicillin with 100 µg/mL streptomycin (Pan-Biotech). Tissue was carefully triturated and centrifuged for 1 min at 20× *g*. Supernatant was cautiously transferred to a new falcon tube and centrifuged for 5 min at 300× *g*. The supernatant was discarded and the pellet resuspended in plating media. Before plating, the T75 flask (Sarstedt, Label A, The Netherlands), coated overnight with 0.1 mg/mL PLL (P2636-100MG, Sigma-Aldrich) in *ortho*-boric acid (VWR chemicals) buffer (pH 8.5), was rinsed three times in MilliQ water. The dissociated glial cells were filtered through a 70 µm cell strainer (Corning, VWR, Darmstadt, Germany) and plated into the T75 flasks. After 2–3 days, the culture medium was changed and further maintained in 37 °C at 5% CO_2_. After 10–12 DIV, microglia were separated from astrocytes by agitation (230 rpm) at 37 °C for 3–4 h for obtaining the microglia. Adherent astrocytes were harvested using 0.25% trypsin-EDTA for 5 min in an incubator. Astrocytes were collected and centrifuged at 473× *g* for 10 min. The cell pellet was resuspended in plating media consisting of 45.5% DMEM (high glucose with L-glutamine, 4500 mg/L D-glucose, 110 mg/L sodium pyruvate, Gibco, Thermo Fisher Scientific), 45.5% HAMS-F12 nutrient mix (Gibco, Thermo Fisher Scientific), 8% FBS (Pan-Biotech), 1% Pen/Strep (Pan-Biotech), and seeded into 60 mm-diameter Petri dishes (Nunclon^TM^ Delta surface, Thermo Fisher Scientific).

### 2.4. Aβ Treatment

Human synthetic amyloid-beta 42 (Aβ_(1–42)_, Tocris Cat. No. 1191, Bio-Techne GmbH, Wiesbaden, Germany) and human synthetic amyloid-beta 40 (Aβ_(1–40)_, Tocris Cat. No 1428, Bio-Techne GmbH) were reconstituted in dimethylsulfoxide (DMSO, (≥99.5%, Carl Roth, Karlsruhe, Germany)) to a 250 µM stock concentration, sonicated for 10 min, and centrifuged at 13,523× *g* at 16 °C [30]. Finally, the 12 DIV hippocampal primary neurones and glial cells were treated with or without 1 µM of Aβ_(1–40)_ and Aβ_(1–42)_ for 3 h and 12 h. 

### 2.5. Synaptic Study

#### Immunofluorescence

After 3 h and 12 h DMSO and Aβ treatment, the conditioned medium was removed and the cultured hippocampal neurones were washed once with warm phosphate buffer saline (PBS (1X), Gibco, Thermo Fisher Scientific), followed by fixation with 4% paraformaldehyde (PFA, Merck Millipore, Darmstadt, Germany) in PBS (1X) containing 15% D-(+)-saccharose (≥99.7%, Carl Roth) at 4 °C for 10 min. After fixation, the cells were washed three times with PBS (1X) before being permeabilised with 0.1% triton X-100 (Carl Roth) for three minutes at room temperature (RT). The neurones were then washed three times with PBS (1X) for 10 min while gently shaking on a shaker at RT. The neurones were blocked in PBS (1X) containing 10% foetal bovine serum (FBS, Pan Biotech) and 1% normal goat serum (NGS, Vector laboratories, Biozol, Germany) for 1 h on a shaker at RT. Afterward, they were incubated with two primary antibodies: guinea-pig anti-vesicular glutamate transporter-1 (VGlut-1 (1:200), Synaptic Systems, 135304, Göttingen, Germany) and mouse anti-postsynaptic density protein-95 (PSD-95 (1:1000), clone (7E3-1B8), Thermo Fisher Scientific, MA1-046) overnight on a shaker at 5 °C. After washing the neurones three times with PBS they were incubated with two secondary antibodies: goat anti-mouse Alexa Fluor 488 (1:1500, Molecular probes) and goat anti-guinea-pig cyanine fluorescent dye (Cy3) (1:1000, Jackson Immuno Research, Cambridgeshire, UK) on a shaker for 1.5 h at RT. Nuclear staining was performed using DAPI (1:2000). All primary and secondary antibodies, as well as DAPI (Carl Roth), were diluted in PBS (1X) containing 5% FBS and 1% NGS. The coverslips with neurones were mounted on glass slides (Duran 76 mm × 26 mm, Carl Roth) with Immu-mount^®^ (Shandon, Thermo Fisher). Immunofluorescent images for quantification of synaptic profiling between pre-and postsynaptic markers were captured using an Olympus IX83 invert microscope with DP80 camera (Olympus, Shinjuku, Japan) using the UPlanSApo 100×/1.4 oil objective with the following filter modules: U-F39002 AT-FITC for Alexa Fluor 488, U-F39004 AT-CY3 for Cy3, and U-FF for DAPI. Background correction, adjustment of brightness and contrast, and selection of regions of interest (ROI) were performed by CellSense software (Olympus, Shinjuku, Japan). Fluorescent images (Figure 1A) were captured with an inverted laser-scanning confocal microscope (SP8, Leica, Wetzlar, Germany) with a 63×/1.4 oil objective (zoom 3) using the Leica software. Images were taken using 488 nm and 564 nm lasers. Background correction, brightness, and contrast were adjusted by ImageJ software (NIH, Bethesda, MD, USA). Further processing of the images was carried out using Adobe Illustrator CC 2020.

### 2.6. Lipidomic Study

#### 2.6.1. Samples Collection for Lipid Analysis

Hippocampal primary neurones (12 DIV) and glial cells were harvested and collected in Eppendorf^®^ LoBind micro-centrifuge tubes (Eppendorf, Omnilab) after 3 h and 12 h with and without Aβ_(1–40)_ and Aβ_(1–42)_ treatment. The supernatant was discarded. Dishes were washed once with cold PBS (1X), neurones and glial cells collected in LoBind tubes after 2 min’ centrifugation (9391× *g*) at 4 °C, the supernatant was discarded, and the pellets were snap-frozen and stored at −80 °C.

#### 2.6.2. Lipid Extraction

Hippocampal primary neurone (12 DIV) and glial-cell pellets were transferred by glass pipettes into transparent glass centrifuge tubes (Corning) containing chloroform (CHCl_3_, SupraSolv^®^, Merck, Darmstadt, Germany), methanol (LiChrosolv^®^ HPLC gradient grade, Merck), and fuming hydrochloric acid 37% (Rotipuran^®^, p.a., ACS, ISO, Carl Roth) solutions supplemented with 1% butylhydroxytoluol (BHT, ≥99%, Carl Roth) and fractioned by pipetting up and down while the sample was on ice. Then, 1 µL TopFluor lysophosphatidic acid (LPA) (702.58 g/mol, Avanti 810280P-1MG), was added per 1 million cells. The TopFluor LPA was dissolved in 1 mL CHCl_3_ (1 mg/mL). Then CHCl_3_ and water (H_2_O, Rotisolv^®^ HPLC gradient grade, Carl Roth) were added, followed by vortexing and 30 min incubation in the dark at RT. Samples were centrifuged at 1260× *g* for 10 min at RT. Finally, the lipid phase was collected using a glass Pasteur pipette into a new glass vial (Thermo Fisher Scientific) and placed in a nitrogen chamber with 6% O_2_ concentration, overnight in the dark. Until measurement, the samples were stored at −20 °C.

#### 2.6.3. High-Performance Thin-Layer Chromatography

We used various external standards as references, such as 1,2-dioleoyl-*sn*-glycero-3-phosphocholine (18:1 (Δ9-Cis) PC (DOPC), Avanti 850375); 1,2-dioleoyl-*sn*-glycero-3-phosphoethanolamine (18:1 (Δ9-Cis) PE (DOPE), Avanti 850725); 1,2-dioleoyl-*sn*-glycero-3-phospho-L-serine (18:1 PS (DOPS), Avanti 840035) for lipid classes of glycerophospholipids, and N-nervonoyl-D-*erythro*-sphingosylphosphorylcholine (24:1 SM, Avanti 860593) for lipid class of sphingolipids. Extracted lipid samples from the neuronal cell lysates were analysed by dilution in 12 µL CHCl_3_ for 1 million cells, using a Hamilton^®^ syringe. The samples and external standard were vortexed and transferred by a Hamilton^®^ syringe to a glass vial with glass micro-insert (31 × 6 mm, Omnilab). The diluted samples and standards were sprayed band-shaped onto the silica gel plates (TLC silica gel 60F_254_ (20 × 10 cm plates), Merck) using the CAMAG automatic TLC sampler 4 (ATS 4). The sprayed bands were separated by capillary force, based on a polarity gradient in the mobile phase, using the CAMAG horizontal developing chamber. The mobile phase contained CHCl_3_, methanol, H_2_O, and ammonia (NH_3_, 32% HiPerSolv^®^ Chromanorm for HPLC, VWR). We performed visualisation (CAMAG TLC Visualizer 2) and scanning (CAMAG TLC Scanner 4) using the mercury lamp at a wavelength of 366 nm for the measurement of the fluorescent signal of the internal standard, Topfluor LPA, followed by a development step with a solution containing copper (II)-sulphate pentahydrate (≥99.5%, p.a., ACS, ISO, Carl Roth), H_2_O, *ortho*-phosphoric acid (85%, Carl Roth), and methanol using the CAMAG Derivatizer. After derivatisation, the plate was placed for 1 h at 120 °C. Finally, the plates were visualised and scanned using a Tungsten lamp with a wavelength of 420 nm. All data were processed by the CAMAG VisionCATS software 2.4, Adobe Photoshop CC 2020, and Adobe Illustrator CC 2020.

#### 2.6.4. Tandem Mass Spectrometry

Lipid analyses were performed using lipid chromatography coupled to tandem mass spectrometry (tandem MS). Lipids were extracted as previously described [31]. Briefly, cells were resuspended in 1 mL H_2_O and directly transferred into glass centrifuge tubes. After addition of 10 µL of the internal standard (30 µM of C17-lysophosphatidylcholine (C17-LPC), C15-ceramide (C15-Cer), C17-sphingosine (C17-So), C34:0 phosphatidylcholine (PC 17:0/17:0), C17-sphingomyelin (C17-SM), C34:0 phosphatidylserine (PS 17:0/17:0), C17-lysophosphatidylserine (C17-LPS), C17-lysophosphatidylethanolamine (C17-LPE), 10 µM C17-sphingosine 1-phosphate (C17-So1P), all from Avanti Polar Lipids, Alabaster, AL, USA, and 1 mM ergosterol from Merck), 200 µL 6 N HCl, 1 mL methanol and 2 mL CHCl_3_ were also added. Samples were vigorously vortexed for 10 min. After centrifugation at 1900× *g*, the lower CHCl_3_ phase was collected. Extraction was repeated with an additional 2 mL of CHCl_3_, and the two CHCl_3_ phases were combined and evaporated using the SpeedVac RVC 2–25 CDplus (Christ, Osterode, Germany). Samples were resuspended in 100 µL methanol: CHCl_3_ (4:1 *v*/*v*) and analysed using the Prominence high-performance liquid chromatography (HPLC) system (Shimadzu, Duisburg, Germany) with a 60 mm × 2 mm MultoHigh 100 RP 18 column with 3 μm particle size (CS-Chromatographie Service, Langerwehe, Germany), which was maintained at 50 °C. The mobile phase A consisted of 1% (*v*/*v*) formic acid in ddH_2_O, and the mobile phase B of 100% methanol. The column was equilibrated in 10% B with a flow rate of 0.5 mL min^−1^. The mobile phase switched to 100% B after sample injection. The flow rate increased linearly from 0.5 mL min^−1^ at 5 min to 1.0 mL min^−1^ at 7 min and remained constant until 10 min. Subsequently, the mobile phase changed to 10% B, and the flow rate decreased linearly from 1.0 mL min^−1^ at 10 min to 0.5 mL min^−1^ at 10.5 min and remained constant until the end of the program at 11.3 min. Detection took place between 2 and 10 min. The injection volume per sample was 10 µL, and samples were cooled to 4 °C. The HPLC system was coupled to the API 2000 triple-quadrupole mass spectrometer (Sciex, Foster City, CA, USA) equipped either with an ESI or an APCI source (Appendix A), both operating in positive mode under the following source parameters: source temperature 450 °C, curtain gas 40, collision gas “low”, ion spray voltage 5500, ion source gas 1 60 (APCI: 30), ion source gas 2 30 (APCI: 60). The analytical results were quantified with Analyst 1.6.2 (AB Sciex, Forster City, CA, USA) based on internal standard samples and an external standard curve.

### 2.7. Data Analysis

For quantification of our synaptic study, we counted active synapses on hippocampal neurones from 12 DIV. Synapses in hippocampal neurones were stained with antibodies specific for synaptic proteins (PSD-95, postsynaptic in green; Vglut-1, presynaptic in red). Only puncta with obvious PSD-95/Vglut-1 overlap (in yellow) were counted as active synapses [32]. We analysed three regions of interest (ROI = 47 µm^2^) on each neurone for an average of ≥12 neurones from four different cultures (*n* = 4) after both 3 h and 12 h of Aβ treatment. In the 3 h treatment, the following average amount of neurones were counted in each group: CTRL (*n* = 12), DMSO (*n* = 9), Aβ_(1–40)_ (*n* = 11) and Aβ_(1–42)_ (*n* = 9). For the 12 h treatment, the following average amount of neurones were counted in each group: CTRL (*n* = 14), DMSO (*n* = 13), Aβ_(1–40)_ (*n* = 13) and Aβ_(1–42)_ (*n* = 17). Statistical analysis was performed using GraphPad Prism 7 (GraphPad Software, San Diego, CA, USA). Values were analysed for normal distribution using the Shapiro–Wilk test. As the data were not normally distributed, they were analysed by performing a Kruskal–Wallis test followed by a post hoc Dunn’s multiple comparison test. All data are presented as mean + standard deviation (SD) and considered to be significant if *p* < 0.05 (*** *p* < 0.0001). Tandem mass spectrometry data of three independent experiments (*n* = 3) from neurones and glial cells were further processed by GraphPad Prism 7, performing logarithmic 10 transformation, and plotted to our negative control (DMSO). This was because the raw data showed that a low concentration of DMSO (final concentration 0.03%) showed an effect on both the lipid composition of glial cells and hippocampal primary neurones, as previously demonstrated [33]. Z-score transformation was performed, and data were plotted as heat maps for each lipid class. 

## 3. Results

### 3.1. Synaptic Loss after Aβ_(1–40)_ and Aβ_(1–42)_ Treatment

To examine whether Aβ species have a preferential early or late effect on the loss of active synapses, primary hippocampal neurones at 12 DIV were treated with (1 µM) Aβ_(1–40)_ or (1 µM) Aβ_(1–42)_ for 3 h and 12 h. Active synapses were analysed by colocalisation of fluorescent markers recognizing vesicular glutamate transport 1 (Vglut-1) and postsynaptic density protein-95 (PSD-95) (Figure 1A–C). We quantified three ROI (47 µm^2^) on the dendritic tree of the hippocampal neurones by examining the colocalisation dots (Figure 1A, white boxes) of a pre- and postsynaptic marker. No significant effect on active synapses was observed after 3 h treatment between our control groups (CTRL and DMSO) compared to Aβ-treated groups (Aβ_(1–40)_ and Aβ_(1–42)_). After 12 h, however, Aβ-treated groups showed a significant reduction in active synapses as compared to the control groups (Figure 1A and Table 1). Both 3 h and 12 h DMSO treatments showed a small decrease in active synapses compared to the CTRL, but this was not significant (Figure 1B,C and Table 1). We observed no significant difference after 3 h and 12 h treatment between Aβ_(1–40)_ and Aβ_(1–42)_ treatments (Figure 1B,C and Table 1). However, 3 h Aβ_(1–40)_ and Aβ_(1–42)_ treatment showed a slight decrease in active synapse numbers as compared to our control groups, but this was not significant (Figure 1B,C and Table 1).

### 3.2. Alterations in Cellular Lipids of Hippocampal Neurones and Glial Cells after Aβ_(1–40)_ and Aβ_(1–42)_ Treatment

#### 3.2.1. Disruption in Cellular Lipids of Hippocampal Neurones and Glial Cells after Aβ_(1–40)_ and Aβ_(1–42)_ Treatment Using High-Performance Thin-Layer Chromatography (HPTLC)

Hippocampal neurones at 12 DIV and glial cells were treated with Aβ_(1–40)_ and Aβ_(1–42)_ for 3 h and 12 h to test whether Aβ species influence cellular lipids, whether that has a preference for a specific class of lipids, and to examine whether Aβ_(1–40)_ and Aβ_(1–42)_ have early effects on the lipid changes in these cells. Two different sets of analyses were performed: HPTLC and tandem MS. Lipids of hippocampal neurones and glial cells were extracted equally from our control groups (CTRL and DMSO) and Aβ-treated groups (Aβ_(1–40)_ and Aβ_(1–42)_) after 3 h and 12 h of treatment. TopFluor lysophosphatidic acid (LPA) was used as the internal standard, to see whether the same amounts of lipids were extracted from each sample of control- and Aβ-treated groups (Appendix A). To analyse which different lipid classes (e.g., sphingolipids and glycerophospholipids) were dysregulated, we showed HPTLC-derivatised copper (II)-sulphate images and their corresponding scanning profiles for hippocampal neurones and glial cells treated with Aβ_(1–40)_ and Aβ_(1–42)_ for 3 h and 12 h (Appendix A). These images and profiles revealed changes in different lipid classes (e.g., sphingolipids and glycerophospholipids), which were in accordance with the external standards used (Appendix A, purple arrows and boxes). Each numbered purple arrow and box refers to external standards that could be detected from hippocampal neurones and glial cells after 3 h and 12 h of Aβ_(1–40)_ and Aβ_(1–42)_ treatment. The copper (II)-sulphate scanning profiles are shown to facilitate examination of the changes between control groups and Aβ-treated groups. The represented retention factor (Rf, Appendix A) and arbitrary unit (AU, Appendix A) are both based on the peak’s end in the copper (II)-sulphate scanning profiles. These show the changes in band intensity of the HPTLC copper (II)-sulphate images and correspond to the numbered labelled purple arrows and boxes (Appendix A). Our data showed small changes in sphingolipids and glycerophospholipids after 3 h Aβ_(1–40)_ and Aβ_(1–42)_ treatment for both neurones (Appendix A) and glial cells (Appendix A), and large changes after 12 h of Aβ_(1–40)_ and Aβ_(1–42)_ treatment (Appendix A). 

#### 3.2.2. Specific Identification of Various Lipid Isoforms after Aβ_(1–40)_ and Aβ_(1–42)_ Treatment from Hippocampal Neurones and Glial Cells Using Tandem Mass Spectrometry Analysis

To corroborate whether Aβ_(1–40)_ and Aβ_(1–42)_ preferentially influence specific sphingolipid or glycerophospholipids isoforms, we treated hippocampal neurones at DIV 12 and glial cells for 3 h and 12 h with Aβ_(1–40)_ and Aβ_(1–42)_ and subsequently analysed them by tandem MS. Here, the characterisation of altered lipid isoforms for hippocampal neurones and glial cells after 3 h and 12 h Aβ_(1–40)_ and Aβ_(1–42)_ treatment are shown, focussing on sphingolipids such as ceramides (Cer), dihydroceramides (DHCer), lactosylceramides (LacCer), monohexosylceramides (HexCer), sphingosine (So), sphinganine (Sa), sphingosine-1-phosphate (Sa1P), sphinganine-1-phosphate (So1P), sphingosylphosphorylcholine (SPC), sphingomylines (SM), dihydrosphingomylines (DHSM), and glycerophospholipids such as phosphatidylcholine (PC), lyso-phosphatidylcholine (LPC), lyso-phosphatidylethanolamine (LPE), lyso-phosphatidylglycerol (LPG), lyso-phosphatidylserine (LPS), Lyso-platelet-activating factor (Lyso-PAF) (Figure 2, Figure 3 and Figure 4). All results are presented as a log_10_ transformation and were performed in triplicate. Furthermore, ratios between the Aβ-treated groups and our negative control (DMSO) were calculated, setting the negative control (DMSO) to 0. After taking the ratios, the Z-score was taken for the total data set. Changes in lipid isoforms are shown as tendencies. This means that the red colours refer to an increase of lipid quantities and the green colour indicates a reduction compared to our negative control (DMSO) which was set to 0 as baseline parameter (Figure 2, Figure 3 and Figure 4).

##### Ceramides (Cer)

Both hippocampal neurones and glial cells showed alterations in Cer isoforms after 3 h and 12 h Aβ_(1–40)_ and Aβ_(1–42)_ treatment (Figure 2). Neurones showed a reduction in C14 Cer after 3 h Aβ_(1–40)_ treatment. An increase of C14 Cer was observed after 3 h and 12 h of Aβ_(1–40)_ and Aβ_(1–42)_ treatment. Other ceramide isoforms (C16 Cer, C18:0 Cer, C20 Cer, C22 Cer, C24:1 Cer) showed an increase in 3 h Aβ_(1–40)_-and 12 h Aβ_(1–42)_-treated neurons. Aβ-treated glial cells showed a strong increase after 3 h Aβ_(1–40)_ and Aβ_(1–42)_ treatment for C14 Cer. After 12 h, Aβ_(1–42)_-treated glial cells showed a strong reduction compared to Aβ_(1–40)_ for C14 Cer. A weak reduction in C18:1 Cer for 3 h Aβ_(1–42)_ and 12 h Aβ_(1–40)_ (Figure 2) was also observed. Other ceramide isoforms (C16 Cer, C18:0 Cer, C20 Cer, C22 Cer, C24:1 Cer) showed a weak increase after 12 h Aβ_(1–42)_-treatment on glial cells. An increase was observed in C24 Cer after 3 h Aβ_(1–40)_ as well as after 3 h and 12 h Aβ_(1–42)_ in treated neurones. No changes were observed in C24 Cer of Aβ_(1–40)_- and Aβ_(1–42)_-treated glial cells (Figure 2).

##### Dihydroceramides (DHCer)

In Aβ_(1–42)_-treated glial cells, C14 DHCer, C16 DHCer and C18:0 DHCer decreased after 3 h and increased after 12 h treatment. However, in glial cells C14 DHCer and C18:0 DHCer was increased after 3 h of Aβ_(1–40)_ treatment. After 3 h Aβ_(1–40)_- and Aβ_(1–42)_-treatment on glial cells, a reduction in C20 DHCer and an increase in C22 DHCer was observed. After 12 h there was a decrease in Aβ_(1–40)_- and an slight increase in Aβ_(1–42)_-treated glial cells. C24:1 DHCer showed a weak reduction after 3 h Aβ_(1–40)_ and Aβ_(1–42)_ treatment as well as an increase after 12 h Aβ_(1–42)_ treatment in glial cells. After 3 h and 12 h Aβ treatment, glial cells showed a decrease in C24 DHCer (Figure 2). Aβ_(1–40)_- and Aβ_(1–42)_-treated hippocampal neurones showed a specific decrease in C20 DHCer after 3 h, and no changes after 12 h Aβ_(1–40)_ and Aβ_(1–42)_ treatment. C22 DHCer showed a decrease in 3 h Aβ_(1–40)_- and 12 h Aβ_(1–42)_-treated hippocampal neurones. An increase in C22 DHCer was seen in 3 h Aβ_(1–42)_- and 12 h Aβ_(1–40)_-treated hippocampal neurones (Figure 2). Other Dihydroceramides isoforms (C14 DHCer, C16 DHCer, C24:1 DHCer, C24 DHCer) showed a weak increase in 3 h Aβ_(1–40)_-treated neurons. However, 3 h Aβ_(1–42)_ treatment showed a reduction in C14 DHCer, C16 DHCer, C24:1 DHCer, and increase in C24 DHCer (Figure 2). 

##### Lactosylceramides (LacCer)

Glial cells treated with Aβ_(1–40)_ and Aβ_(1–42)_ showed a reduction in all LacCer isoforms after 12 h. In 3 h Aβ_(1–42)_-and 12 h Aβ_(1–40)_-treated glial cells, decreases in C18:0 LacCer and C22 LacCer and an increase after 12 h Aβ_(1–42)_ treatment were seen. In C16 LacCer and C24:1 LacCer an increase after 12 h Aβ_(1–40)_ and Aβ_(1–42)_ treatment on glial cells was seen. A reduction was observed in C16 LacCer after 3 h Aβ_(1–40)_ and a decrease after 3 h Aβ_(1–42)_ treatment on glial cells. After 3 h, Aβ_(1–40)_-treated hippocampal neurones showed an increase in C18:0 LacCer. A decrease in C16 LacCer and C18 LacCer was seen in 3 h Aβ_(1–42)_-treated hippocampal neurons (Figure 2). A slight decrease was observed in C18:0 LacCer after 12 h Aβ_(1–40)_ and Aβ_(1–42)_-treatment on neurones as well as in C22 LacCer after 3 h Aβ_(1–42)_ treatment.

##### Monohexosylceramides (HexCer)

After Aβ_(1–40)_ treatment, C16 HexCer showed a strong reduction in 12 h-treated hippocampal neurones and in 3 h-treated glial cells, whereas 3 h Aβ_(1–42)_-treated glial cells and 3 h Aβ_(1–40)_- and Aβ_(1–42)_-treated hippocampal neurones showed an increase (Figure 2). C22 HexCer and C24 HexCer showed strong increases after 12 h Aβ_(1–40)_ and decreases in 12 h Aβ_(1–42)_ for glial cells and a slight reduction in 12 h Aβ_(1–40)_- and a strong increase in 12 h Aβ_(1–42)_-treated hippocampal neurones. After 3 h of Aβ_(1–40)_- and Aβ_(1–42)_-treated hippocampal neurons and glial cells, a strong increase was seen in C22 HexCer. C24 HexCer showed an increase in 3 h Aβ_(1–40)_- and Aβ_(1–42)_-treated glial cells as well as in 3 h Aβ_(1–42)_-treated hippocampal neurons. Slight increases were observed in C14 HexCer, C18:0 HexCer, and C20 HexCer after 3 h Aβ_(1–40)_ treatment on hippocampal neurones. However, C14 HexCer showed an weak increase after 3 h Aβ_(1–40)_ and 12 h Aβ_(1–42)_ treatment on hippocampal neurones. A slight reduction was seen in C14 DHCer after 3 h Aβ_(1–42)_ and 12 h Aβ_(1–40)_ treatment on glial cells. C18:0 HexCer showed an increase after 3 h Aβ_(1–40)_ and 12 h Aβ_(1–42)_ on hippocampal neurones. A slight decrease in C18:0 HexCer was observed after 3 h Aβ_(1–42)_ treatment (Figure 2).

##### Sphingosine (So) and Sphingosine-1-Phosphate (So1P)

Hippocampal neurones and glial cells showed weak changes of d18:1 So after both 3 h and 12 h Aβ_(1–40)_ and Aβ_(1–42)_ treatments (Figure 3). There was a decrease in d18:1 So1P after 3 h Aβ_(1–42)_ treatment in hippocampal neurones and an increase after 12 h Aβ_(1–42)_ treatment in glial cells. A reduction in d18:1 So1P was also observed after 3 h and 12 h Aβ_(1–40)_ treatment of hippocampal neurons (Figure 3). An increase in d18:1 So1P for both 3 h Aβ_(1–40)_-treated glial cells and 12 h Aβ_(1–42)_-treated hippocampal neurones was observed (Figure 3).

##### Sphinganine (Sa) and Sphinganine-1-Phosphate (Sa1P)

A prominent reduction was seen in d18:0 Sa for glial cells after 3 h and 12 h Aβ_(1–40)_ and Aβ_(1–42)_ treatment (Figure 3). In hippocampal neurones d18:0 Sa was slightly increased after 3 h Aβ_(1–40)_ treatment and slightly reduced after 12 h Aβ_(1–42)_ treatment. A reduction was also observed in d18:0 Sa1 P after 3 h Aβ_(1–40)_- and Aβ_(1–42)_-treated hippocampal neurones, as well as 3 h and 12 h Aβ_(1–40)_-treated glial cells. No strong reduction was observed in d18:0 Sa1P after 12 h Aβ_(1–40)_ and Aβ_(1–42)_ treatment of hippocampal neurones as well as in 3 h Aβ_(1–42)_-treated glial cells (Figure 3). 

##### Sphingosylphosphorylcholine (SPC)

Both, hippocampal neurones and glial cells showed alterations in SPC 16:0 after 3 h Aβ_(1–40)_ and Aβ_(1–42)_ treatment. The most prominent decrease in SPC 16:0 was shown after 3 h of Aβ_(1–40)_ and Aβ_(1–42)_ treatment for both hippocampal neurones and glial cells. A decrease was also observed after 12 h of Aβ_(1–42)_ treatment for both hippocampal neurones and glial cells (Figure 3).

##### Sphingomyelines (SM)

All SM isoforms (C14 SM, C16 SM, C18:1 SM, C18:0 SM, C20 SM, C22 SM, C24:1 SM, C24 SM, C26:1 SM, and C26 SM) showed a slight increase after 3 h Aβ_(1–40)_ and Aβ_(1–42)_ treatment on hippocampal neurones, and glial cells after 3 h and 12 h Aβ_(1–40)_ and Aβ_(1–42)_ treatment (Figure 3). The most prominent reductions were observed in C26 SM for glial cells after 3 h Aβ_(1–40)_ and 12 h Aβ_(1–42)_ treatment and in C26:1 SM for hippocampal neurones after 3 h and 12 h Aβ_(1–40)_ and Aβ_(1–42)_ treatment. 3 h and 12 h Aβ_(1–40)_-treated hippocampal neurones showed a strong increase in C26 SM (Figure 3).

##### Dihydrosphingomyelines (DHSM)

Almost all DHSM isoforms (C14 DHSM, C16 DHSM, C18:1 DHSM, C18:0 DHSM, C20 DHSM, C22 DHSM, C24:1 DHSM, C24 DHSM, C26:1 DHSM and C26 DHSM) showed an increase after 3 h Aβ_(1–40)_ and Aβ_(1–42)_ and 12 h Aβ_(1–40)_ treatment, as well as a decrease after 12 h Aβ_(1–42)_ treatment of hippocampal neurones. The most prominent reduction in DHSM for hippocampal neurones was observed after 3 h in C14 DHSM after Aβ_(1–40)_ treatment and C26 DHSM after Aβ_(1–42)_ treatment, and for all DHSM isoforms after 12 h Aβ_(1–42)_ treatment (Figure 3). However, the most prominent increase in DHSM for hippocampal neurones was observed after 3 h in C14 DHSM after Aβ_(1–42)_ treatment and C26 DHSM after Aβ_(1–40)_ treatment, and for all DHSM isoforms after 12 h Aβ_(1–40)_ treatment (Figure 3). Almost all DHSM isoforms (C14 DHSM, C16 DHSM, C18:1 DHSM, C18:0 DHSM, C20 DHSM, C22 DHSM, C24:1 DHSM, C24 DHSM, C26:1 DHSM, and C26 DHSM) from glial cells treated with Aβ were increased, whereas in glial cells, the most prominent increase are observed after 3 h Aβ_(1–40)_ treatment for C14 DHSM and after 12 h Aβ_(1–42)_ treatment for C18:0 DHSM, C20 DHSM, and C22 DHSM. Additionally, glial cells showed a prominent decrease in C14 DHSM after 12 h Aβ_(1–42)_ (Figure 3).

##### Phosphatidylcholines (PC) and Lyso-Phosphatidylcholines (LPC)

A prominent decrease in PC (38:0) in hippocampal neurones after 3 h Aβ_(1–40)_ as well as 12 h Aβ_(1–40)_ and Aβ_(1–42)_ was observed, as well as an strong increase after 3 h Aβ_(1–42)_ treatment. All other PC isoforms (PC (28:0), PC (30:0), PC (30:1), PC (32:0), PC (32:1), PC (34:1), PC (34:2), PC (36:1), PC (36:2), PC (36:2), PC (36:3), PC (38:0), PC (38:1), PC (38:2), PC (38:3), PC (38:4)) showed a decrease in 3 h Aβ_(1–40)_- and 12 h Aβ_(1–40)_-treated glial cells. Beside PC (28:0), a weak increase was observed after 12 h Aβ_(1–42)_ treatment in glial cells. The following PC isoforms (PC (28:0), PC (30:1), PC (32:0), PC (32:1), PC (34:1), PC (36:1), PC (38:0), PC (38:1), PC (38:2), PC (38:3)) were reduced after 3 h Aβ_(1–40)_ treatment of glial cells. Glial cells also showed a prominent increase in PC (36:0) after 3 h Aβ_(1–40)_ and Aβ_(1–42)_ treatment. No changes were observed in PC (38:0) after 12 h Aβ_(1–42)_ treatment of glial cells. LPC (20:0) showed a strong increase after 3 h Aβ_(1–40)_ treatment of hippocampal neurones and in 3 h Aβ_(1–42)_- and 12 h Aβ_(1–40)_-treated glial cells (Figure 4). A prominent decrease was observed in LPC (20:0) in 3 h Aβ_(1–40)_ and 12 h Aβ_(1–42)_ treated glial cells and a weak decrease in 12 h Aβ_(1–40)_-treated hippocampal neurones (Figure 4).

##### Lyso-Phosphatidylethanolamine (LPE)

A reduction in all LPE isoforms (LPE (16:0), LPE (16:1), LPE (18:0), LPE (18:1), LPE (18:2), and LPE (20:4)) was shown in 3 h Aβ_(1–40)_- and Aβ_(1–42)_-treated hippocampal neurones. A slight decrease in all LPE isoforms (LPE (16:0), LPE (16:1), LPE (18:0), LPE (18:1), LPE (18:2), and LPE (20:4)) was shown in 3 h Aβ_(1–42)_-treated glial cells (Figure 4). An increase was observed in all LPE isoforms (LPE (16:0), LPE (16:1), LPE (18:0), LPE (18:1), LPE (18:2), and LPE (20:4)) in 12 h Aβ_(1–42)_-treated glial cells (Figure 4). 12 h Aβ_(1–40)_-treated hippocampal neurones showed an decrease in LPE (16:1), LPE (18:0), LPE (18:2), and LPE (20:4) as well as an increase in LPE (16:0), LPE (18:1) isoforms. An increase of LPE (16:0), LPE (18:1), and LPE (20:4)) was seen in 12 h Aβ_(1–42)_-treated hippocampal neurones. 

##### Lyso-Phosphatidylglycerol (LPG)

LPG (14:1) and LPG (16:1) showed a reduction in 12 h Aβ_(1–40)_-treated hippocampal neurones. Glial cells showed a strong increase after 3 h Aβ_(1–40)_ and Aβ_(1–42)_ treatment in LPG (14:1) and after 12 h Aβ_(1–40)_ treatment in LPG (16:1). In addition, LPG (16:1) showed a decrease after 3 h Aβ_(1–40)_ and Aβ_(1–42)_ treatment of glial cells (Figure 4). 

##### Lyso-Phosphatidylserine (LPS)

LPS (18:2) showed an increase in 3 h Aβ_(1–40)_- and Aβ_(1–42)_-treated glial cells as well as in 3 h and 12 h Aβ_(1–40)_-treated hippocampal neurones. A reduction was observed in Aβ_(1–42)_-treated hippocampal neurons after 3 h and 12 h (Figure 4). A decrease was observed in LPS (18:0) after 3 h Aβ_(1–40)_- and Aβ_(1–42)_-treated glial cells. LPS (16:0) was reduced in 3 h Aβ_(1–40)_-treated hippocampal neurones, as well as in 12 h Aβ_(1–40)_-treated glial cells (Figure 4).

##### Lyso-Platelet-Activating Factor (Lyso-PAF)

Lyso-PAF showed a reduction in 12 h Aβ_(1–40)_-treated glial cells, as well as in hippocampal neurones after 3 h and 12 h, for both Aβ_(1–40)_ and Aβ_(1–42)_ treatment. A prominent increase of lyso-PAF was observed in 3 h Aβ_(1–40)_- and Aβ_(1–42)_-treated glial cells (Figure 4).

## 4. Discussion

In this study, using different quantitative cellular techniques in different Aβ-treated primary brain cells, we showed for the first time that lipid profiles change after treatment and before synaptic loss was observed. We showed that human Aβ species exogenously applied to primary neurones and glial cells influence the numbers of active synapses and led to lipid alterations in a time-dependent manner. We also demonstrated that Aβ_(1–40)_ and Aβ_(1–42)_ treatment have a specific-species dependent influence on the integrity of cellular lipids in hippocampal neurones and glial cells. 

Aβ_(1–40)_ and Aβ_(1–42)_ induces synaptic loss only after 12 h of treatment. 

Synaptic dysfunctions due to Aβ accumulation are strongly associated with the cognitive disturbances of AD. The Aβ species Aβ_(1–40)_ and Aβ_(1–42)_ are the major forms of amyloid β peptides in the brain_,_ whereby Aβ_(1–42)_ seems to be more toxic than Aβ_(1–40)_ [30,34,35]. We found no selective influences of Aβ_(1–40)_ or Aβ_(1–42)_ on the loss of active synapses in cultured hippocampal neurones. Moreover, both Aβ species induced synaptic loss after a 12 h treatment, whereas a 3 h treatment did not show effects on synapses either with Aβ_(1–40)_ or Aβ_(1–42)_. Differences between these Aβ species on the synapse alterations are dependent on the concentration and aggregated forms of Aβ [30]. Moreover, Fu et al. (2017) [35] showed that the Aβ_(1–42)_ oligomers have a moderate but significantly higher level of neurotoxicity, whereas monomers have the weakest neurotoxic effect (~20%) on neuronal cells, without differences between Aβ_(1–40)_ and Aβ_(1–42)_. These results correspond to our findings on primary hippocampal neurones. 

Aβ_(1–40)_ and Aβ_(1–42)_ differently influence cellular lipids before synaptic loss appears.

Previous studies have shown changes in different lipid classes and a correlation to AD pathology. Although we do not fully understand the connection between AD and lipid metabolism, there is more and more evidence that lipids could be a useful blood biomarker in the diagnostics of AD, in addition to other risk factors [36,37]. Nevertheless, it is poorly understood which lipid isoforms are dysregulated on a cellular level. We showed here that changes in lipid composition in hippocampal neurones and glial cells occur early, even before synaptic loss becomes evident, and independently of the Aβ species. 

Our data indicate an increase of ceramides (especially C14 Cer) in glial cells and a reduction of these in hippocampal neurones. Both types of brain cells indicate an early reduction of specific dihydroceramides (C20 DHCer and C22 DHCer). These different amounts of lipids caused by Aβ could be a result of activated apoptotic-survival pathways and inflammatory activity due to their response to oxidative stress [38]. 

Glycosylated ceramides exhibited a reduced (C16 HexCer; C20 HexCer) or an increased level (C16 HexCer, C22 HexCer, C24 HexCer, C16 LacCer, C18 LacCer, C24:1 LacCer) for either hippocampal neurones or glial cells. Sphingomyelin (C26 SM) was upregulated in hippocampal neurones and downregulated in glial cells. It has been suggested that Aβ_(1–40)_, Aβ_(1–42)_ and Aβ-islet amyloid polypeptide concentrations lead to nSMase-mediated induction of apoptosis, resulting in neuronal cell death and Aβ-induced oxidative stress. Aβ_(1–40)_ and Aβ_(1–42)_ species and their oligomeric isoforms modify ceramide levels, which could explain the elevated nSMase and reduced cellular sphingomyelin activity [23,24,39,40,41]. In comparison with Aβ_(1–40)_, it was shown that Aβ_(1–42)_ directly activates nSMase by a γ-secretase-dependent mechanism, leading to increased ceramide levels [42]. 

Furthermore, we found an upregulation of d18:1 So1P in hippocampal neurones and glial cells after 3 h and 12 h treatment. An increase in So1P levels has been discussed in relation to Aβ-induced toxicity in hippocampal neurones and glial cells in vitro and in vivo [43,44]. We also showed a reduction of SPC (16:0), Sa (d18:0 Sa), and Sa1P (d18:0 Sa1P) in hippocampal neurones and glial cells after Aβ_(1–40)_ and Aβ_(1–42)_ treatment. These reductions might be regulated by the increase of toxic Aβ species’ aggregates by enhancing β-secretase mediated pathways and the anti-inflammatory response in neuronal cells [45,46]. Furthermore, after Aβ_(1–40)_ and Aβ_(1–42)_ treatment we showed an early increase in specific isoforms of phospholipids in hippocampal neurones and lyso-phospholipids in hippocampal neurones and glial cells. Additionally, we verified that Aβ-treated hippocampal neurones and glial cells showed a specific reduction in phospholipids (PC (38:0), LPC (20:0), all LPE, LPG, LPE (18:1), LPS (18:2) and Lyso-PAF) as well as a specific upregulation in phospholipids (PC (38:0), LPC (20:0), all LPE, LPG, LPE (18:1), LPS (18:2), and Lyso-PAF) after either 3 h or 12 h Aβ treatment. These glycerophospholipid alterations have been discussed in association with cytosolic phospholipase A2 hyperactivity in neuro-inflammatory responses and synapse-mediated Aβ toxicity [31,47]. 

In conclusion, we found that after Aβ_(1–40)_ or Aβ_(1–42)_ treatment, levels of specific sphingolipid and glycerophospholipid isoforms change before there is any detectable loss of active synapses. No significant differences were seen between the effects of Aβ_(1–40)_ and Aβ_(1–42)_ species in the loss of active synapses. Our data contributes to a better understanding of the cellular influences of Aβ species. It remains to be elucidated whether these influences of Aβ species on changes in membrane lipid profiles could be useful as an early diagnostic biomarker in amyloidosis-related disorders.

## Figures and Tables

**Figure 1 ijms-23-02300-f001:**
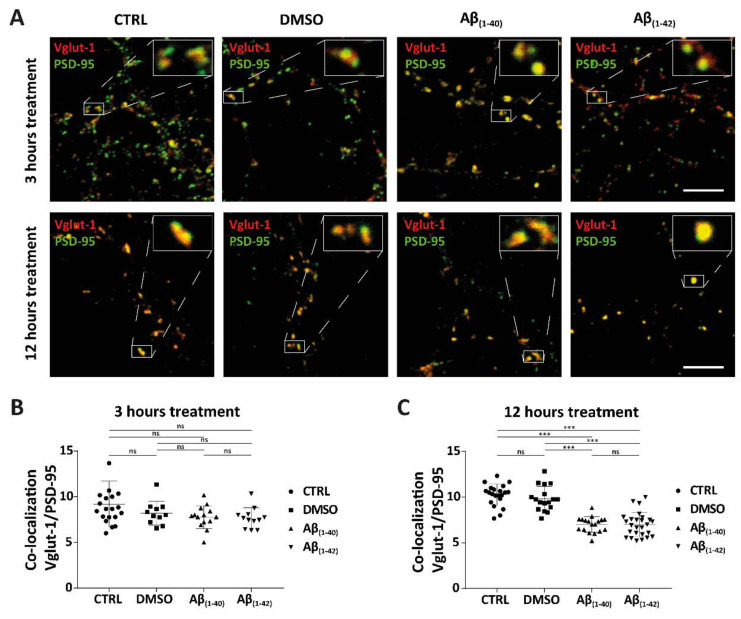
Synaptic loss of active synapses of Aβ accumulation in hippocampal neurones. (**A**–**C**) The synaptic study showed the colocalisation of presynaptic (Vglut-1) and postsynaptic (PSD-95) markers, which was defined as active synapses, in hippocampal neurones at DIV12 for both control groups (CTRL and DMSO) and Aβ-treated groups ((1 µM) Aβ_(1–40)_ and (1 µM) Aβ_(1–42)_) at different time points (3 h and 12 h) of treatment (all groups from four independent cultures (*n* = 4)). In the 3 h treatment, the following average number of neurones was counted in each group: CTRL (*n* = 12), DMSO (*n* = 9), Aβ_(1–40)_ (*n* = 11) and Aβ_(1–42)_ (*n* = 9). In the 12 h treatment, the following average number of neurones was counted in each group: CTRL (*n* = 14), DMSO (*n* = 13), Aβ_(1–40)_ (*n* = 13), and Aβ_(1–42)_ (*n* = 17). For both 3 h and 12 h treatments, 3 ROI from each neurone was counted. (**A**) Representative confocal images of control groups and Aβ-treated groups of hippocampal neurones stained for both presynaptic (Vglut-1) and postsynaptic markers (PSD-95) are shown. There was a reduction in active synapses after 12 h of Aβ_(1–40)-_ and Aβ_(1–42)_-treated neurones compared to control groups. White boxes show enlargements of an example of defined active synapses for both control groups and Aβ-treated groups (3 h and 12 h). (**B**,**C**) Quantification of active synapses from control- and Aβ_-_treated groups of hippocampal neurones. (**B**) No significant effect on active synapses was seen in Aβ-treated groups after 3 h (3 ROI of average counted cells; *n* = 10, from four independent cultures (*n* = 4)). (**C**) A significant reduction of active synapses was observed after 12 h of Aβ_(1–40)_- and Aβ_(1–42)_-treated neurones compared to control groups. No significance (ns) was seen between CTRL and DMSO, nor between Aβ_(1–40)_- and Aβ_(1–42)_-treated neurones (3 ROI of average counted cells; *n* = 14, from four independent cultures (*n* = 4)). *** *p* < 0.0001; scale bars represent 5 µm. The size of the ROI was 47 µm^2^.

**Figure 2 ijms-23-02300-f002:**
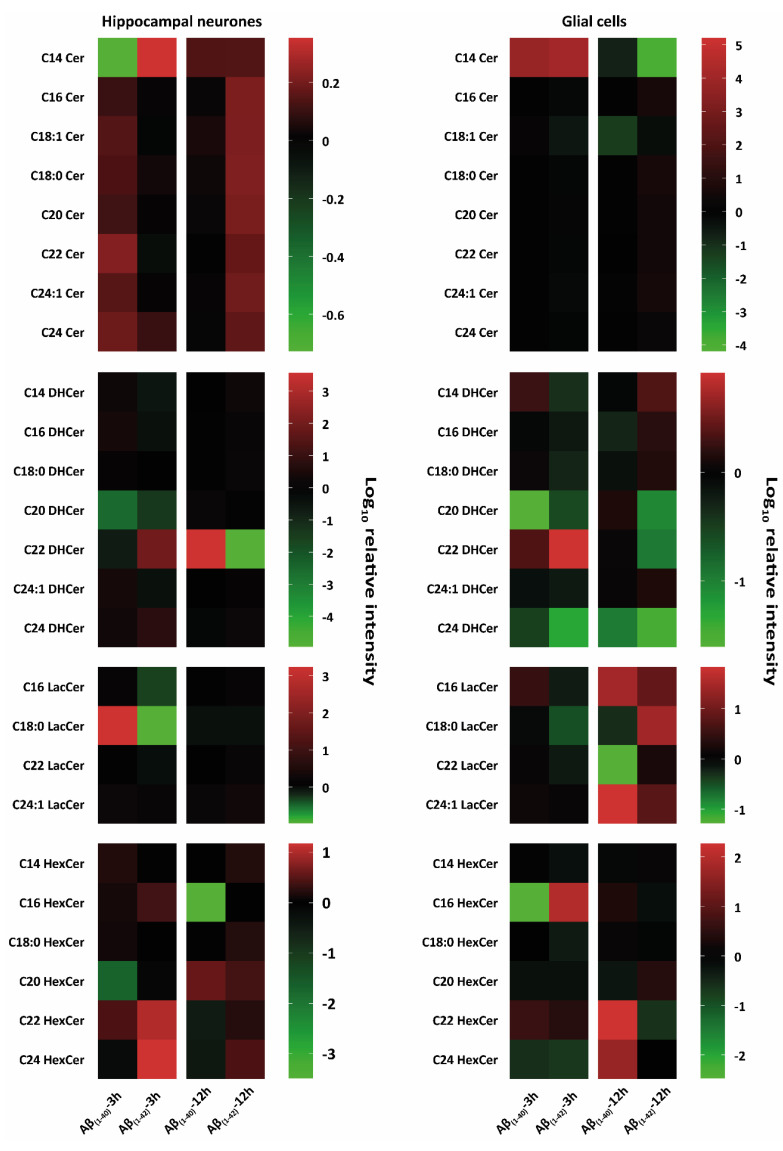
Heat-map-based mass spectrometry analysis of sphingolipid classes from hippocampal neurones and glial cells after 3 h and 12 h Aβ treatment. Here, we examined different sphingolipid isoforms: ceramides (Cer), dihydroceramides (DHCer), lactosylceramides (LacCer), and monohexosylceramides (HexCer) after 3 h and 12 h treatment with (1 µM) Aβ_(1–40)_ and (1 µM) Aβ_(1–42)_ of hippocampal neurons at DIV 12 and glial cells (all groups, from three independent experiments (*n* = 3)). Changes of these lipid classes are shown as logarithmic (log_10_) relative intensity (arbitrary unit); the green colour refers to a reduction, and the red colour refers to an increase of lipid levels compared to our negative control (DMSO), which was set to 0 as baseline. Both hippocampal neurones and glial cells showed an increased and reduced intensity of Cer, DHCer, LacCer, and HexCer isoforms after both 3 h and 12 h Aβ_(1–40)_ and Aβ_(1–42)_ treatment. Different changes in the lipid classes are shown between treated hippocampal neurones and glial cells. Aβ-treated hippocampal neurones showed prominent increases and decreases in Cer isoforms and LacCer and HexCer isoforms after 3 h and 12 h of Aβ_(1–40)_ and Aβ_(1–42)_ treatment. There was both a prominent intensity increase and a decrease of DHCer isoforms after 3 h and 12 h of Aβ_(1–40)_ and Aβ_(1–42)_ treatment. Glial cells showed prominent lipid changes of Cer, DHCer, LacCer, and HexCer isoforms after 3 h and 12 h of Aβ_(1–40)_ or Aβ_(1–42)_ treatment.

**Figure 3 ijms-23-02300-f003:**
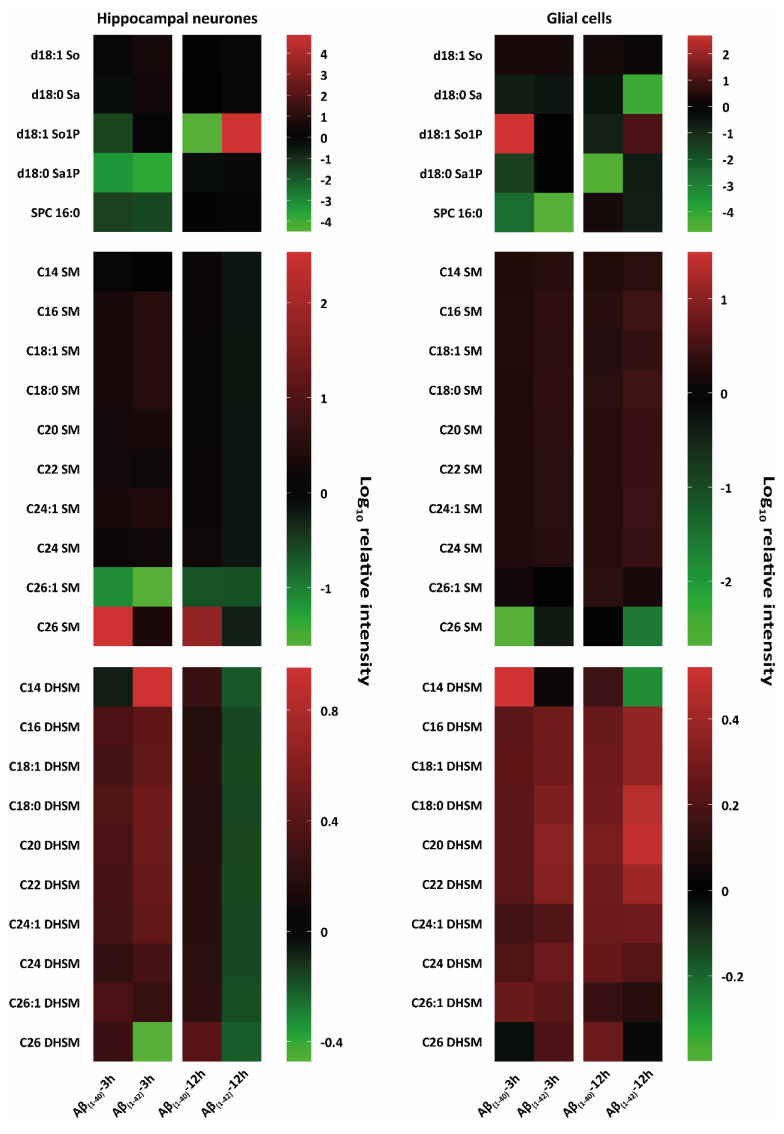
Heat-map-based mass spectrometry analysis of sphingophospholipid classes from hippocampal neurones and glial cells after 3 h and 12 h Aβ treatment. Here, we examined different sphingophospholipid isoforms; sphingosine (d18:1 So), sphinganine (d18:0 Sa), sphingosine-1-phosphate (d18:1 So1P), sphinganine-1-phosphate (d18:1 Sa1P), sphingomyelines (SM) and dihydrosphingomyelines (DHSM) after 3 h and 12 h treatment with (1 µM) Aβ_(1–40)_ and (1 µM) Aβ_(1–42)_ hippocampal neurones at DIV 12 and glial cells (all groups, from three independent experiments (*n* = 3)). Changes of these lipid classes are shown as logarithmic (log_10_) relative intensity (arbitrary unit); the green colour refers to a reduction and the red colour refers to an increase in lipid levels compared to our negative control (DMSO), which was set to 0 as baseline. Major intensity changes in lipid classes of d18:1 So, d18:0 Sa, d18:1 So1P, d18:1 Sa1P after 3 h and 12 h Aβ_(1–40)_ and Aβ_(1–42)_ treatment of hippocampal neurones and glial cells were observed. Specific lipid intensity changes after Aβ_(1–40)_ and Aβ_(1–42)_ treatment of hippocampal neurones of very long fatty acid SM compared to Aβ-treated glial cells were detected.

**Figure 4 ijms-23-02300-f004:**
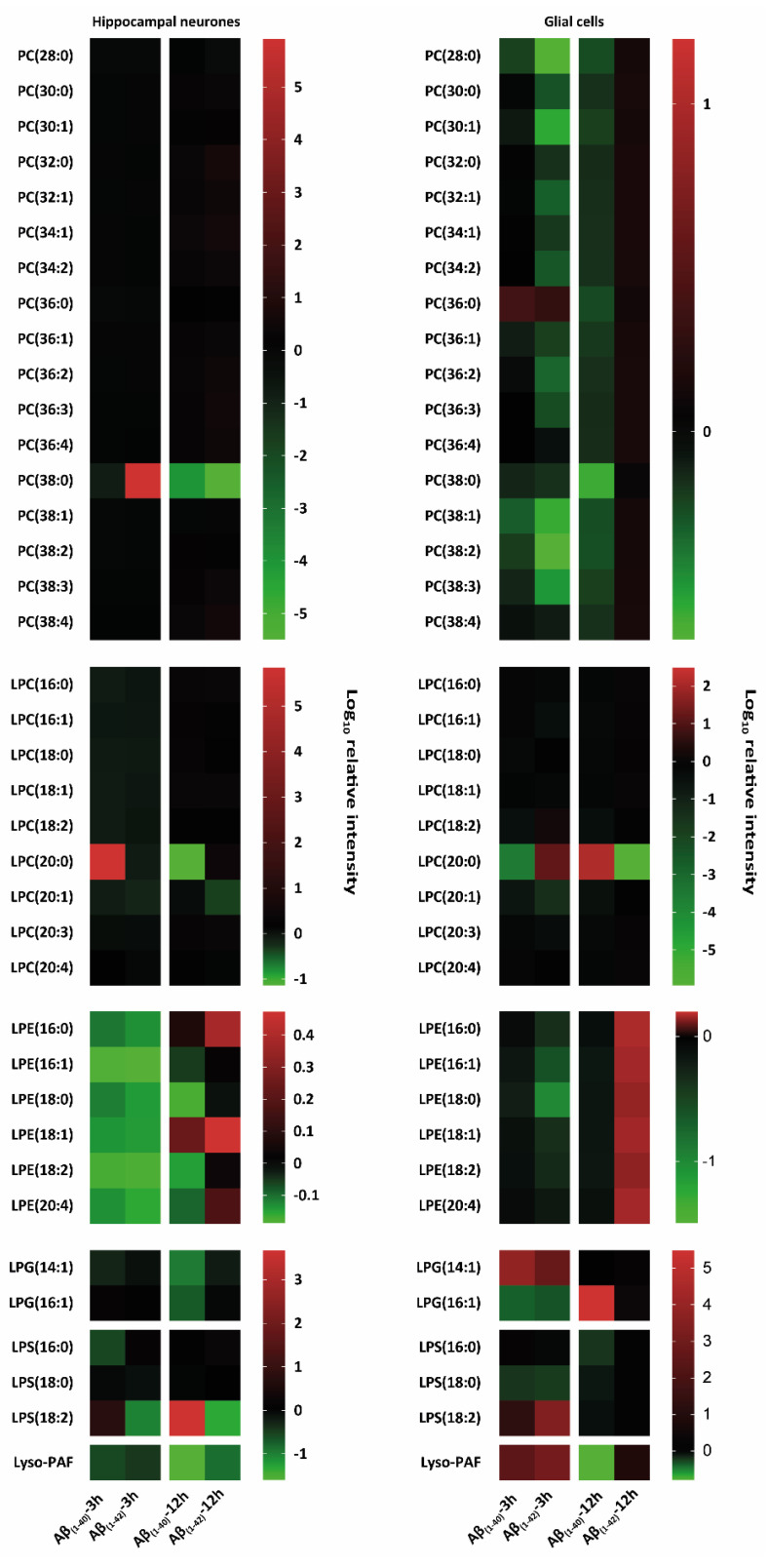
Heat map-based mass spectrometry analysis of glycerophospholipid classes from hippocampal neurones and glial cells after 3 h and 12 h Aβ treatment. Here, we examined different glycerophospholipids isoforms: phosphatidylcholine (PC), lyso-phosphatidylcholine (LPC), lyso-phosphatidylethanolamine (LPE), lyso-phosphatidylglycerol (LPG), lyso-phosphatidylserine (LPS), and lyso-platelet-activating factor (Lyso-PAF) after 3 h and 12 h treatment with (1 µM) Aβ_(1–40)_ and (1 µM) Aβ_(1–42)_ of hippocampal neurones at DIV 12 and glial cells (all groups, from three independent experiments (*n* = 3)). Changes in these lipid classes are shown as logarithmic (log_10_) relative intensity (arbitrary unit); the green colour refers to a reduction and the red colour refers to an increase in lipid levels compared to our negative control (DMSO), which was set to 0 as baseline. Hippocampal neurones specifically showed PC (38:0) intensity changes after 3 h and 12 h Aβ_(1–40)_ and Aβ_(1–42)_ treatment compared to an overall reduction in PC for Aβ_(1–40)_ and Aβ_(1–42)_-treated glial cells. LPC, LPG, LPS, and Lyso-PAF showed specific changes after 3 h and 12 h Aβ_(1–40)_ and Aβ_(1–42)_ treatment in both hippocampal neurones and glial cells.

**Table 1 ijms-23-02300-t001:** List of the statistically adjusted *p*-value (from Figure 1B,C) between separate groups (CTRL, DMSO, Aβ_(1–40)_ and Aβ_(1–42)_) after 3 h and 12 h of treatment conditions. A Kruskal–Wallis test was performed, followed by a Dunn’s multiple comparison test.

	3 h Treatment	12 h Treatment
Kruskal–Wallis test	0.0732	<0.0001
Dunn’s multiple comparison test	adjusted *p*-value	adjusted *p*-value
CTRL vs. DMSO	>0.9999	0.6106
CTRL vs. Aβ_(1–40)_	0.3017	<0.0001
CTRL vs. Aβ_(1–42)_	0.0867	<0.0001
DMSO vs. Aβ_(1–40)_	>0.9999	<0.0001
DMSO vs. Aβ_(1–42)_	>0.9999	<0.0001
Aβ_(1–40)_ vs. Aβ_(1–42)_	>0.9999	>0.9999

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
