# Peer review of "Aβ-Induced Alterations in Membrane Lipids Occur before Synaptic Loss Appears"

_ijms, 2022, doi:10.3390/ijms23042300_

Round 1

Reviewer 1 Report

The authors performed Lipidomic study in cultured hippocampal neuron and glia cells to detect the changes of lipid metabolism with Abeta treatment. Overall , the expriments are well designed and presented. The conclusion that specific lipid class sush as sphingolipid and glycerphospholipid is changed after abeta treatment is clear.  I have suggested publish this manuscript as a resource paper in IJMS and the authors can further detect the lipid in microglia.

Author Response

We thank the reviewer for the positive feedback and are very happy to take the suggestion for the further experiments.

Reviewer 2 Report

The manuscript titled "Aβ-induced alterations in membrane lipids occur before synaptic loss appears" demonstrated that amyloid-beta (Aβ) treatment reduces the number of active synapses and alters lipidomes including multiple sphingolipids. 

The manuscript was written clearly and I agree that the manuscript provides novel information on neurons' response in lipidome to Aβ series. I have a few technical questions that could augment the quality of the manuscript if addressed. 

  1. The authors argued in the first data that the active synapse is defined as the colocalization of pre- and post-synaptic markers. However, the definition of colocalization is not provided. The figure 1's representative colocalization images all show partial colocalization, thus it is difficult to define colocalization versus non-colocalization. A clear definition for determining colocalization and representative images should be provided in the manuscript. Also, the number of independent experiments (not neuron numbers) is provided. While these technical questions can be trivial but very important for assessing the reproducibility of the work. 
  2. In the same subject, the second dataset also does not have the number of independent experiments/samples. In rigorous lipidomic experiments, a minimum of five independently-performed experiments are required, considering the variability of the lipidome in general. Please provide the number of independent experiments for the lipidomic data, and if the number is not enough, please provide more datasets. 

Author Response

  1. The authors argued in the first data that the active synapse is defined as the colocalization of pre- and post-synaptic markers. However, the definition of colocalization is not provided. The figure 1's representative colocalization images all show partial colocalization, thus it is difficult to define colocalization versus non-colocalization. A clear definition for determining colocalization and representative images should be provided in the manuscript. Also, the number of independent experiments (not neuron numbers) is provided. While these technical questions can be trivial but very important for assessing the reproducibility of the work. 

We agree with the reviewer, that number of experiments and numbers of neurons per experiments were not explained detailed enough and therefore specified this in the “Material and Methods” part and the figure legends. We defined N as the total number of independent experiments and n as the total number of neurons used for statistic.

We also followed the reviewer’s advice and included a definition for determining colocalization in the “Material and Methods” part 2.7. Data Analysis.

  1. In the same subject, the second dataset also does not have the number of independent experiments/samples. In rigorous lipidomic experiments, a minimum of five independently-performed experiments are required, considering the variability of the lipidome in general. Please provide the number of independent experiments for the lipidomic data, and if the number is not enough, please provide more datasets. 

We agree with the reviewer, that the total number of independent experiments is important. In our study we used three independent experiments for each group. We included these numbers in the “Material and Methods” part and the figure legends. We believe that for this first study the number of experiments shows a valid result. However, we also agree with the reviewer that an increase in the number of experiments in the lipid analysis is very important once we will go into the detailed analysis of individual lipid classes. We will follow this advice of the reviewer and increase the number of independent experiments to at least five in our next detailed studies. We thank you for this important advice.